# Color Confinement and Bose-Einstein Condensation

Masanori Hanada[b], Hidehiko Shimada[e] and Nico Wintergerst[c]

[b]*Department of Mathematics, University of Surrey, Guildford, Surrey, GU2 7XH, UK*

[e]*Mathematical and Theoretical Physics Unit, Okinawa Institute of Science and Technology,*
*1919-1 Tancha, Onna-son, Okinawa 904-0495 Japan*
*Yukawa Institute for Theoretical Physics, Kyoto University*
*Kitashirakawa Oiwakecho, Sakyo-ku, Kyoto 606-8502 Japan*

[c]*The Niels Bohr Institute, University of Copenhagen,*
*Blegdamsvej 17, 2100 Copenhagen Ø, Denmark*

## Abstract

We propose a unified description of two important phenomena: color confinement in large-$N$ gauge theory, and Bose-Einstein condensation (BEC). We focus on the confinement/deconfinement transition characterized by the increase of the entropy from $N^0$ to $N^2$, which persists in the weak coupling region. Indistinguishability associated with the symmetry group — SU($N$) or O($N$) in gauge theory, and S$_N$ permutations in the system of identical bosons — is crucial for the formation of the condensed (confined) phase. We relate standard criteria, based on off-diagonal long range order (ODLRO) for BEC and the Polyakov loop for gauge theory. The constant offset of the distribution of the phases of the Polyakov loop corresponds to ODLRO, and gives the order parameter for the partially-(de)confined phase at finite coupling. We demonstrate this explicitly for several quantum mechanical systems (i.e., theories at small or zero spatial volume) at weak coupling, and argue that this mechanism extends to large volume and/or strong coupling. This viewpoint may have implications for confinement at finite $N$, and for quantum gravity via gauge/gravity duality.

MH would like to dedicate this paper to Keitaro Nagata.

# 1 Introduction

In this paper we point out a hitherto unnoticed connection between two important phase transitions: Bose-Einstein condensation (BEC) [1] and the confinement/deconfinement transition [2, 3] in large-$N$ gauge theories.

Throughout this paper, we adopt a characterization of confinement and deconfinement by the increase of energy and entropy from order $N^0$ to $N^2$ (for fields in the adjoint representation), or to $N^1$ (for fields in the fundamental representation). There are two important motivations to consider the confinement/deconfinement transition in this sense. First, in the context of gauge/gravity duality, a detailed understanding of this transition may provide important insight on how information about the spacetime geometry is encoded on the gauge theory side. The confined and deconfined phases are considered to describe the vacuum and black hole geometry on the gravity side, respectively [4]. Second, this characterization may lead to a new way of understanding the more traditional 'dynamical' confinement [5] defined in terms of the existence of a linear potential between probe quarks. This approach was pioneered in Refs. [6, 7] in which gauge theories defined on a spatial sphere are considered. It was shown that the confinement characterized by the $N$-dependence of the free energy carries over to weak or even vanishing coupling, and small

or vanishing spatial volume, where the system and the transition can be described analytically.[1] For finite interaction and large volume, the confinement defined by the two characterizations are expected to coincide, as argued e.g. in [8].[2]

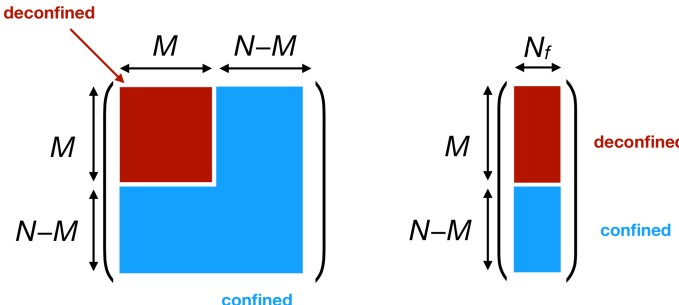

Figure 1: Partial confinement in the gauge sector with adjoint matter (left) and vector matter (right). The elements shown in blue are confined, whereas the elements shown in red are deconfined. These figures are taken from Ref. [11]. Partial confinement can be defined in a gauge-invariant manner. For details, see Sec. 2.1, around Eq. (1).

The purpose of this paper is twofold. First, we obtain a better understanding of the mechanism behind the confinement/deconfinement transition in the weakly-coupled regime by exploiting the connection to the physics of BEC. As concrete and solvable examples, we consider SU($N$) Yang-Mills on a small three sphere, the gauged O($N$) vector model on a small two-sphere, and a system of $N$ identical bosons. While all those examples are essentially quantum mechanics rather than quantum field theory,[3] the underlying mechanism we identify is applicable to quantum field theory as well. Second, we consider the problem of extending the understanding at weak-coupling to the strongly-coupling regime. Here, the analogy to BEC provides us with new insight on the order parameters which may be used to discriminate the confined and the deconfined phases for all values of the coupling.

The key concept underlying the connection between confinement and BEC is the idea of *partial confinement* recently introduced in Refs. [12, 13, 14, 11, 15]. In the partially-confined phase[4] of Yang-Mills theory, only the degrees of freedom associated with the

---

[1]For asymptotically free theories, the small volume limit and the weak coupling limit are identical.

[2]In order to extract lessons applicable to the strong coupling regime from weak-coupling calculations, the crucial assumption is that the strong- and weak-coupling regions are smoothly connected without any phase boundary. Whether such a phase boundary is absent or not is model dependent. For some models, the comparisons with strong coupling results obtained via holography and/or lattice simulation gave credible support to the absence of the phase boundary; see e.g. Refs. [9, 10]. For an elaboration on this point, see the discussion section.

[3]For quantum field theory compactified on a small sphere where nonzero-momentum modes can be integrated out, the effective theory of zero-momentum modes is essentially quantum mechanics.

[4]Whether to call this phase partial confinement or partial deconfinement is of course purely a matter of taste. In this paper, we prefer to use the term partial *confinement*, since it parallels the term Bose-Einstein condensation.

$M \times M$ submatrix (red region in Fig. 1) are deconfined, whereas the remaining degrees of freedom (blue region in Fig. 1) are confined.[5] In this sense, the confined and deconfined sectors coexist in the space of colors, and thermodynamic quantities, for example entropy, in the partially-confined phase can be understood as a sum of two terms associated with the two components, i.e. the confined and the deconfined sectors. In Refs. [11, 15], the existence of a partially-confined phase was demonstrated for several weakly-coupled theories by explicitly counting the states contributing to thermodynamic quantities. It was further shown in Refs. [11, 15] that, for weakly coupled theories, the confined degrees of freedom are in their ground state. Thus, as we will elaborate in Sec. 2, the confinement/deconfinement transition has the following characteristic features:

1. The transition occurs even in the weak-coupling limit which can capture important parts of the physics of the transition in strongly-coupled systems.

2. In the (partially) confined phase, a large fraction of the degrees of freedom falls into the ground state (the confined sector).

3. The ground state and the excited states (the deconfined sector) coexist and thermodynamics can be understood from the point of view of a system with these two components.

Furthermore, as we will show in Sec. 3, partial confinement also has the following feature:

4. Positive interference due to the gauge symmetry is the key mechanism of confinement.

The crucial element of our proposal is that precise counterparts of the above features exist for BEC. Namely:

1. The transition occurs even in the weak-coupling limit (the ideal Bose gas) which can capture important parts of the physics of the transition in strongly-coupled systems.

2. In the condensed phase, a large fraction of particles fall into the ground state (the Bose-Einstein condensate).

3. The system consists of particles in the ground state and excited states. The thermodynamic properties can be understood from the point of view of a system with these two components.

4. Positive interference due to the permutation symmetry is the key mechanism of condensation.

A standard example of BEC with non-vanishing interactions is the superfluidity of $^4$He. The confined and deconfined sectors in Yang-Mills theories correspond to super- and

---

[5]How partial confinement is reconciled with gauge invariance is explained in detail in Ref. [11]. See also Sec. 2.1.

normal-fluid components, respectively. One may find it surprising that the weak-coupling calculation of 4d Yang-Mills [6, 7] captures the essence of strongly-coupled dynamics obtained by lattice simulation or holography. From our new point of view, this parallels the fact that a good part of the characteristic features of superfluidity in $^4$He, which is interacting via the van der Waals force, can be understood starting with the free theory, as first pointed out by F. London [16].

The connection between confinement and BEC becomes particularly transparent in a model that is almost tailor-made for this purpose: the gauged O($N$) vector model. In Sec. 2, we show how confinement in this model [17] is related to confinement in Yang-Mills, and to BEC in a system of $N$ identical bosons, focussing on the essential features 1.-3. listed above.

Once appreciating the analogy explained in Sec. 2, it is straightforward to uncover the common mechanism behind confinement and BEC: the indistinguishability (or equivalently, the redundancy) of states due to gauge symmetry or permutation symmetry leads to a parametrically large enhancement of the ground state, known as *Bose enhancement* or *positive interference of the ground-state wave function.* We will explain this mechanism in Sec. 3, and find a precise, quantitative characterization of this interference effect in Sec. 4.

As is well-known, the finite temperature system is described by the Euclidean path integral with the compactified temporal direction. The Polyakov loop $P$ is defined by the gauge covariant path ordered exponential $P = \mathrm{P}e^{\int A_0 dt}$ along a closed path extended in the temporal direction. (The trace of $P$ is also called Polyakov loop.) Since the path ordered exponential is a unitary matrix, its eigenvalues, which are of course gauge invariant, are phase factors of the form $e^{i\theta_j}$ ($j = 1, \cdots, N$). Later we will use the density of these phases $\rho(\theta)$ in the large-$N$ limit. $P$ and $\rho$ can depend on the spatial position of the temporal loop. When used as the order parameter for the confinement/deconfinement transition, usually the spatial average is considered. In this paper we consider the Polyakov loop in the fundamental representation. There is a deep connection to a class of phase transtions characterized by the behavior of the Polyakov loop, first advocated by Gross, Witten [18] and Wadia [19, 20] (the GWW transition). That is, the description of a system of identical bosons as a theory with S$_N$ gauge symmetry permits a straightforward definition of a Polyakov loop, and we show that the formation of a BEC can also be interpreted as a GWW transition.[6] Therefore, both confinement and BEC are characterized by the change of the distribution of the phases of Polyakov loop. We will further show that this characterization of the transition based on the Polyakov loop is closely related to the more traditional characterization based on off-diagonal long range order (ODLRO) [21, 22]. This argument readily generalizes to Yang-Mills models as well. In particular, for an ideal Bose gas, we prove explicitly that ODLRO and the criteria based on the Polyakov loop are equivalent.

In Sec. 5 we discuss possible applications to quantum gravity via holography and con-

---

[6]Here we define the 'GWW transition' by the disappearance of the gap in the distribution of Polyakov loop phases. Other details of the phase transition, including the order of transition, depend on model-specific details such as the dimension and matter content. Note that the original context of the GWW model (arising from the 2D Yang-Mills theory) is not relevant in our discussion.

finement in finite-$N$ theories.

Throughout the paper, we set the Planck constant $\hbar$ and Boltzmann constant $k_{\mathrm{B}}$ to be unity.

# 2   The correspondence in the weak-coupling limit

Let us commence our analysis at zero coupling. First, we provide the necessary background on partial confinement in Yang-Mills theory in Sec. 2.1. The gauged O($N$) vector model is particularly well suited to establish the connection between confinement in Yang-Mills and BEC, and we will introduce it in Sec. 2.2. We will close the section by explaining BEC in the system of identical bosons in Sec. 2.3. These three examples form the basis of the connection between confinement and BEC. The common underlying mechanism will be explained in Sec. 3.

Note that Sec. 2.1 and Sec. 2.2 are based on Ref. [11], and Sec. 2.3 explains well-known established results. We will present the known results in such a way that the unknown connection is revealed.[7]

## 2.1   Yang-Mills theory at weak coupling

As a typical example of the confinement/deconfinement transition in large-$N$ Yang-Mills theory, we consider finite temperature pure Yang-Mills theory defined on the space $\mathrm{S}^3$ with the gauge group SU($N$). We consider the free limit of pure Yang-Mills theory, which is solvable analytically [6, 7]. As we have mentioned in the introduction, and as we will explicitly demonstrate, *partial confinement takes place even in the free limit and the free theory captures important features of the confinement/deconfinement transition.* See e.g. Refs. [9, 10] regarding the resemblance between weak- and strong-coupling regions. Most of the dynamical degrees of freedom can be integrated out, since they become massive due to the compactness and the curvature of $\mathrm{S}^3$. In this way, an effective action for the phases of the Polyakov loop is obtained.[8] This effective action can be solved by using standard matrix-model techniques and the results can be naturally explained in terms of partial confinement, as we now review.

Let us start with a precise definition of the partially-confined sector. In a generic partially-confined state, $M \times M$ degrees of freedom are excited. The remaining degrees of freedom are in the ground state, as shown in Fig. 1. The partially-confined states in the Hilbert space of the theory can be constructed in a manifestly gauge-invariant manner. First, we consider a trivial embedding of SU($M$) into SU($N$), as the upper-left block

---

[7]Some readers may find it helpful to refresh their memory about the basic feature of BEC explained in Sec. 2.3 before reading Sec. 2.1 in order to appreciate the close analogy between the two phenomena.

[8]More precisely, the gauge conditions $\partial^i A_i = 0$ and $\partial_t \alpha = 0$, where $\alpha = \int d^3 x_{\mathrm{S}^3} A_0(x) \times \frac{1}{\mathrm{Vol}_{\mathrm{S}^3}}$, are imposed, and the Polyakov loop is defined by $P = e^{i\beta\alpha}$. For more details, see Ref. [7].

(Fig. 1). All SU($M$)-invariant energy eigenstates $|E; \mathrm{SU}(M)\rangle$ are then obtained by exciting the oscillators associated with the $M \times M$ submatrix components while respecting the $M \times M$ part of the Gauss law constraints. When doing this, we keep all oscillators associated with the remaining $N^2 - M^2$ elements (i.e. the elements shown in blue in Fig. 1) in their ground states. The states thus prepared are invariant under an SU($M$)×SU($N - M$) subgroup of the SU($N$) gauge symmetry. Finally, to construct the fully SU($N$)-invariant state $|E\rangle_{\mathrm{inv}}$ with the same energy, we act with all possible gauge transformation on this state and take the superposition:

$$|E\rangle_{\mathrm{inv}} = \mathcal{N}^{-1/2} \int dU \, \mathcal{U} \left( |E; \mathrm{SU}(M)\rangle \right). \tag{1}$$

Here $\mathcal{U}$ stands for the SU($N$) transformation acting on the states in the Hilbert space, which corresponds to the group element $U \in \mathrm{SU}(N)$.[9] The integral is taken over SU($N$). The normalization factor $\mathcal{N}^{-1/2}$ ensures unit normalization of $|E\rangle_{\mathrm{inv}}$. The mapping from the SU($M$)×SU($N - M$)-invariant states to the SU($N$)-invariant states is one-to-one. This allows one to straightforwardly count all such states and explicitly show that they dominate the thermodynamics [11]. In this way, *a large fraction of the degrees of freedom falls into the ground state* (the confined sector), and the *confined sector and deconfined sector coexist*.

The phase structure of the system is shown in Fig. 2. At $T < T_c$, the system is completely (not partially) confined, i.e. $M = 0$ (the blue line in Fig. 2), whereas at $T > T_c$ the system is completely deconfined, i.e. $M = N$ (the red line in Fig. 2). In the description based on the canonical ensemble (with temperature $T$ varied as a controllable parameter), there is a first order deconfinement phase transition at $T = T_c$. This first-order transition is a result of the coexistence of two sectors (the confined and the deconfined sectors) in equilibrium. As is always the case for a first-order transition resulting from such a phase equilibrium, one can sharpen one's understanding by considering the microcanonical ensemble (with energy $E$ varied as a controllable parameter). [10] In this description, at $T = T_c$, the ensemble can be parametrized by $M$ which increases from 0 to $N$. Equivalently, one can choose a parametrization in terms of the (trace of the) Polyakov loop $P$, which increases from 0 to $\frac{1}{2}$ at $T = T_c$. This parametrization follows naturally from the computation of the effective action [6, 7]. As we shall show below, thermodynamic quantities such as energy, entropy, and the distribution of the phases of the Polyakov loop, can be understood from a single relation between $P$ and $M$ [14, 11],

$$P = \frac{M}{2N}. \tag{2}$$

---

[9]When the states are expressed by acting the creation operators $\hat{a}^\dagger_{\mu,ij}$ to the Fock vacuum, $\left( \mathcal{U} \hat{a}^\dagger_\mu \right)_{ij} = \sum_{k,l=1}^N U_{ik} \hat{a}^\dagger_{\mu,kl} U^{-1}_{lj}$.

[10]In the case of partial confinement, even if the transition is not of first order, the confined and deconfined phases can coexist [14, 11, 15]. This happens for example if one introduce fundamental matter [15]. When the transition is not of first order, there is no need to distinguish the canonical and the microcanonical ensembles.

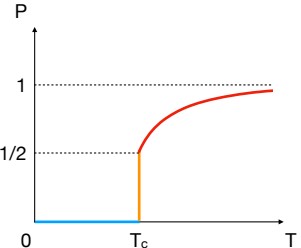
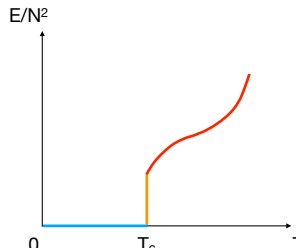

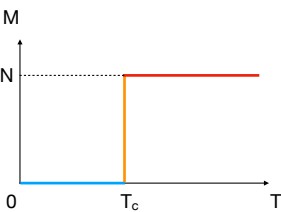

Figure 2: Qualitative features of the phases of the free Yang-Mills on $S^3$ [14, 11]. The Polyakov loop $P$, energy $E$ and the size of the deconfined sector $M$ in weakly-coupled 4d YM on $S^3$ are shown as functions of temperature $T$. The blue, orange and red lines are completely confined, partially confined (equivalently partially deconfined) and completely deconfined phases, respectively.

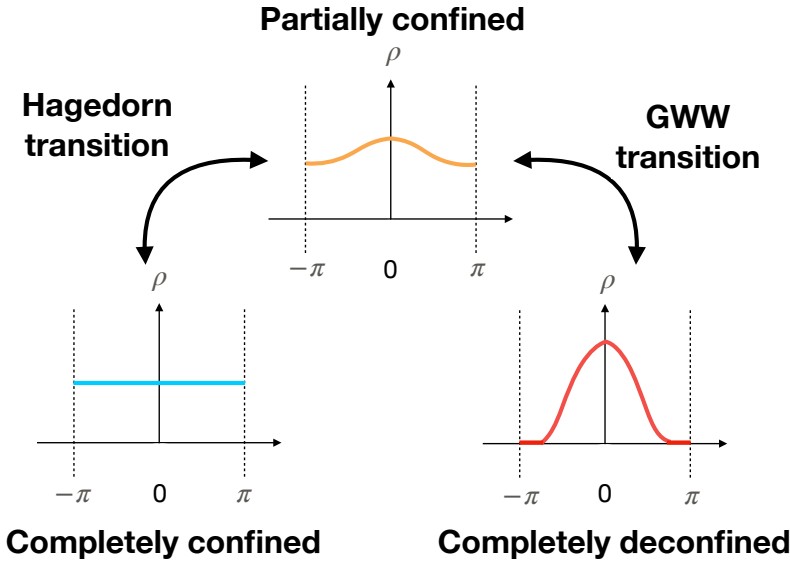

Figure 3: The distribution of the phases of Polyakov loop in the completely confined, partially confined and completely deconfined phases. This figure is taken from Ref. [23].

From the microcanonical viewpoint, therefore, $T = T_c$, $M = 0$ (i.e. $P = 0$) is the transition point from complete to partial confinement, and $T = T_c$, $M = N$ (i.e. $P = \frac{1}{2}$) defines the transition from partial confinement to complete deconfinement. As we will explain below the latter transition is a GWW transition. We denote thermodynamic quantities at this point with the label GWW below.

*This two-component picture of the system explains the thermodynamic properties of the system.* Energy and entropy are given by the sums of those corresponding to the confined and the deconfined sectors. The former is of order $N^0$ and is negligible compared to the latter, which is proportional to $M^2$, since the number of excited degrees of freedom is of order $M^2$. Hence we have

$$E(T = T_c, P; N) = E(T = T_c, P = \frac{1}{2}; N) \times \left(\frac{M}{N}\right)^2 = E_{\text{GWW}}(N) \times |2P|^2 \qquad (3)$$

$$S(T = T_c, P; N) = S(T = T_c, P = \frac{1}{2}; N) \times \left(\frac{M}{N}\right)^2 = S_{\text{GWW}}(N) \times |2P|^2 \qquad (4)$$

where we ignored the zero-point energy. This relation actually holds for the weak-coupling limit of pure Yang-Mills on $S^3$. From these relations, we obtain

$$E(T = T_c, P = M/2N, N) = E_{\text{GWW}}(M), \qquad S(T = T_c, P = M/2N, N) = S_{\text{GWW}}(M),$$
$$(5)$$

where $E_{\text{GWW}}(M)$ and $S_{\text{GWW}}(M)$ are the energy and entropy at the GWW-transition point in the SU($M$) theory. By combining it with the one-to-one mapping (1), we can see that the partially-confined states dominate thermodynamics.

The essence of these relations (5) is as follows [12, 14]. Consider SU($N$)- and SU($N'$)-theories, with $N' < N$ (Fig. 4). Since energy and entropy are dominated by the deconfined sector, there is no apparent difference between the SU($N$)- and SU($N'$)-theories until the size of the deconfined sector $M$ reaches $N'$. (Note that we are assuming the weak-coupling limit here. At finite coupling, all color degrees of freedom can interact with each other, and hence the SU($N$)- and SU($N'$)-theories can behave differently.) Beyond this point, SU($N'$) is completely deconfined, $M$ cannot grow further, and hence the system lies at the GWW-transition point in the SU($N'$)-theory. Therefore, the relations (5) follow naturally.

The distribution of Polyakov loop phases can also be clarified from this point of view. As shown in Fig. 3, in the completely confined phase $T < T_c$ the distribution $\rho(T; N; \theta)$ is constant. For the completely deconfined phase $T > T_c$, $\rho$ becomes 0 at some finite value (i.e., there is a gap). At $T = T_c$, for $0 < P < \frac{1}{2}$, the phase distribution is inhomogenous but has no gap. At $P = \frac{1}{2}$, a gap forms and thus this point is the GWW transition. The distribution $\rho$ can be computed explicitly by the effective action approach [6, 7]. At $T = T_c$, it is given by [6, 7]

$$\rho(T = T_c, P; N; \theta) = \frac{1}{2\pi} (1 + 2P \cos \theta) \qquad (6)$$

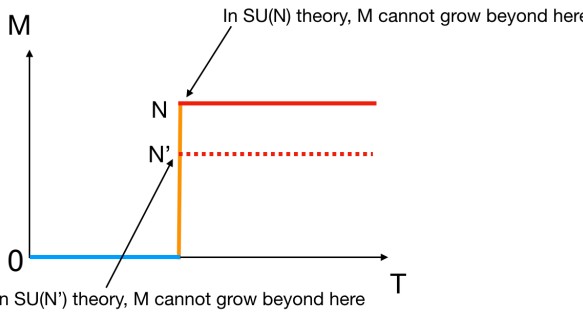

Figure 4: In the weak-coupling limit, the deconfined sector at $M = N'$ corresponds to the GWW-transition point of the SU($N'$)-theory. This figure is taken from Ref. [23].

up to $1/N$ corrections. By using $P = \frac{M}{2N}$, one can rewrite this as

$$
\begin{aligned}
\rho(T = T_c, P; N; \theta) &= \frac{1}{2\pi}\left(1 - \frac{M}{N}\right) + \frac{M}{N} \cdot \frac{1}{2\pi}\left(1 + \cos\theta\right) \\
&= \frac{1}{2\pi}\left(1 - \frac{M}{N}\right) + \frac{M}{N} \cdot \rho_{\mathrm{GWW}}(\theta; M).
\end{aligned} \tag{7}
$$

The first term (which is a constant) and the second term (which becomes zero at $\theta = \pm\pi$) are the respective contributions from the confined and deconfined phases. We see that $M$ phases are in the deconfined sector, while the rest is in the confined sector. Note that $\rho_{\mathrm{GWW}}(\theta; M)$ can have a nontrivial $M$-dependence in general (e.g., when the fundamental quarks are added [15]), though for pure Yang-Mills it is simply $\rho_{\mathrm{GWW}}(\theta; M) = \frac{1+\cos\theta}{2\pi}$.

In this subsection, we have observed the features 1, 2 and 3 mentioned in the introduction; we have studied the free theory and shown that among $N^2$ color degrees of freedom, $N^2 - M^2$ fall into the confined sector, while $M^2$ degrees of freedom are excited. Thermodynamic quantities can be understood from the phase equillibrium between the confined and deconfined sectors. The feature 4 will be explained in Sec. 3 and Sec. 4.

## 2.2 Gauged O($N$) Vector Model at weak coupling

In this section we consider the gauged O($N$) vector model. It is a particularly instructive example to understand the connection between confinement at weak coupling and BEC. As we will explain in this section, due to the gauge-singlet constraint, this model can exhibit the transition from confinement to deconfinement, characterized by the increase of entropy

from $N^0$ to $N^1$ (see also Fig. 5) [17][11]. As we will see below, *partial confinement takes place even in the free limit* [11].

We start with the 3d free theory on the two-sphere of radius $R$, following Ref. [17].[12,] We consider an $N$-component vector of scalar fields $\vec{\phi}(x) = (\phi_1(x), \cdots, \phi_N(x))$ which transforms in the fundamental representation of the $O(N)$ symmetry group and consider its free theory in the $O(N)$-singlet sector. The $N$ components of $\vec{\phi}(x)$ resemble the $N$ bosons that will be discussed in Sec. 2.3, and the gauge symmetry resembles the permutation symmetry.

In its bare essentials, projection onto the singlet sector is achieved by coupling $\vec{\phi}$ to a Lagrange multiplier $\lambda$ via

$$S = \int -\vec{\phi}^T \left( D_t^2 + \partial_i^2 - \frac{1}{4} \right) \vec{\phi},$$

(8)

where the Lagrange multiplier appears inside a gauge covariant derivative $D_t \equiv \partial_t + i\lambda$. In three dimensions, this theory appears in the weak coupling or small radius limit of an interacting conformal field theory. More precisely, one can conformally couple $\vec{\phi}(x)$ to an $O(N)$ gauge field $A_\mu$ with Chern-Simons action; the free singlet theory is then obtained by taking the level to infinity. The Lagrange multiplier appears as the gauge holonomy around the thermal circle that survives the free limit and is in this way connected to the Polyakov loop. For simplicity of discussion, we will from now on only refer to the latter.

At finite temperature, (8) can be studied in exactly the same way as our previous example of $SU(N)$ Yang-Mills following the basic idea and tools introduced by [6, 7]. [13]. Explicitly, one derives an effective action for the phase of the Polyakov loop after integrating out all massive excitations [24, 17]. After minimizing the effective action, the Polyakov loop is zero at $T = 0$, nonzero at any $T > 0$, and the GWW transition, which is the transition to complete deconfinement, takes place at $T_{\text{GWW}}(N) = \frac{\sqrt{3}}{\pi R}\sqrt{N}$.

Below the GWW-temperature, an energy eigenstate can be expressed in the form of eq. (1),

$$|E\rangle_{\text{inv}} = \mathcal{N}^{-1/2} \int dU \, \mathcal{U} \left( |E; O(M)\rangle \right).$$

(9)

Here $|E; O(M)\rangle$ denotes states for which $\phi_{M+1}, \cdots, \phi_N$ are in the ground state, as shown in the right panel of Fig. 1. In this way, *a large fraction of the degrees of freedom falls into the ground state* (the confined sector), and the *confined sector and deconfined sector coexist*.

In order to show the above more explicitly and to see how temperature and $M$ are related, let us look at the Polyakov loop closely. By using $b = \frac{TR}{\sqrt{N}}$, and taking $b$ to be of order one, the distribution of the Polyakov loop phase $\theta$ is written as

$$\rho(\theta) = \frac{1}{2\pi} + \frac{2b^2}{\pi} f(\theta),$$

(10)

---

[11]This scaling holds in an appropriate double scaling limit involving the radius and $N$. See the end of this section for details.

[12]For simplicity we set the number of flavor $N_f$ in Ref. [17] to be one.

[13]Since the gauge group is $SO(N)$ rather than $U(N)$, the density $\rho(\theta)$ necessarily becomes symmetric under the refelection $\theta \leftrightarrow -\theta$.

where

$$f(\theta) = -\frac{\pi^2}{12} + \frac{(|\theta| - \pi)^2}{4}. \tag{11}$$

At $b = b_{\mathrm{GWW}} = \frac{\sqrt{3}}{\pi}$, the GWW transition takes place; the distribution becomes zero at $\theta = \pm\pi$.

We can rewrite $\rho(\theta)$ as

$$\rho(\theta) = \left(1 - \frac{b^2}{b_{\mathrm{GWW}}^2}\right) \cdot \rho_{\mathrm{confine}}(\theta) + \frac{b^2}{b_{\mathrm{GWW}}^2} \cdot \rho_{\mathrm{GWW}}(\theta), \tag{12}$$

where $\rho_{\mathrm{GWW}}(\theta)$ is the distribution of the phases at $b = b_{\mathrm{GWW}}$, and $\rho_{\mathrm{confine}}(\theta) = \frac{1}{2\pi}$ is the distribution of the phases in the confined phase. The parameter $b$ is related to the size of the deconfined sector $M$ as [11] (see Fig. 5)

$$\frac{M}{N} = \frac{b^2}{b_{\mathrm{GWW}}^2}. \tag{13}$$

Equivalently,

$$TR = b\sqrt{N} = b_{\mathrm{GWW}}\sqrt{M}. \tag{14}$$

Note that this is the critical temperature of the $O(M)$ theory: $T_{\mathrm{GWW}}(M) = b_{\mathrm{GWW}}\sqrt{M}$. Therefore, the identification leads to

$$\rho(\theta, T = T_{\mathrm{GWW}}(M)) = \frac{1}{2\pi}\left(1 - \frac{M}{N}\right) + \frac{M}{N} \cdot \rho_{\mathrm{GWW}}(\theta; M). \tag{15}$$

This relation is analogous to (7). As expected, $M$ phases are in the deconfined sector, while the rest is in the confined sector [14, 11].

At $1 \ll T \leq T_{\mathrm{GWW}}$, the energy scales as $E = AT^5$, with an $N$-independent coefficient $A = 16\,\zeta(5)$, and the entropy is $S = \frac{5}{4}AT^4$. Therefore, at $T = T_{\mathrm{GWW}}(M)$, the following relations hold:

$$E = E_{\mathrm{GWW}}(M), \qquad S = S_{\mathrm{GWW}}(M). \tag{16}$$

These equations are analogous to (5). Namely, energy and entropy, which are dominated by the deconfined sector, can precisely be explained by $O(M)$-partial-confinement. As in the case of free Yang-Mills, the Polyakov loop, energy and entropy are consistently explained by the same $M$ defined by (13) and (14); the $O(M)$-deconfined phase of $O(N)$-model corresponds to the GWW point in the $O(M)$-model. In this way, *the two-component picture of the system explains the thermodynamic properties of the system.*

These calculations can be generalized to $(d + 1)$-dimensional theory on $\mathrm{S}^d$. The critical temperature of the $O(N)$-theory is [24, 17]

$$T_{\mathrm{GWW}}(N) = \left(\frac{N}{4(1 - 2^{1-d})\zeta(d)}\right)^{1/d} \cdot \frac{1}{R}, \tag{17}$$

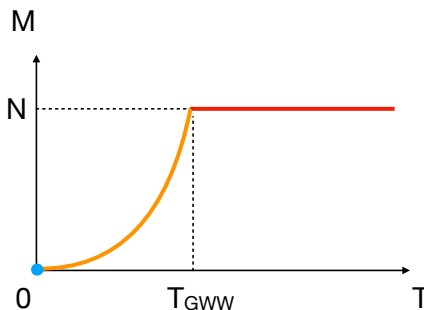

Figure 5: A schematic picture of the size of the deconfined block $M$ as a function of temperature $T$, in the free gauged $O(N)$ vector model on $S^d$. The system is partially confined at $0 < T < T_{\text{GWW}}$.

and the size of the deconfined sector is

$$\frac{M}{N} = \left(\frac{T}{T_{\text{GWW}}}\right)^d. \tag{18}$$

The key relations (15) and (16) remain unchanged.

Again, we have seen the features 1, 2 and 3 mentioned in the introduction; we have studied the free theory, and shown that among $N$ color degrees of freedom, $N - M$ fall into the confined sector, while $M$ degrees of freedom are excited. Once more, feature 4 — the importance of the interference — will be explained in Sec. 3 and Sec. 4.

In Ref. [17], the radius $R$ is taken to be $N$-independent. If instead we take $R = \sqrt{N}$ (or, for generic dimension $d$, $R \propto N^{1/d}$), then $b = \frac{TR}{\sqrt{N}}$ simply equals $T$, and the natural temperature scale becomes $N$-independent. This corresponds to the thermodynamic limit of identical bosons with fixed particle density, which is discussed in Sec. 2.3. In this limit, the free energy, entropy and energy are of order $N$.

## 2.3    BEC of non-interacting particles

The relevance of the BEC of an ideal gas for understanding the superfluidity of $^4$He, which is interacting, was first pointed out by F. London [16]. This idea was elaborated to a two-component fluid theory, corresponding to particles in the ground and excited states,

respectively, which gave a remarkably good phenomenological understanding of the super-fluidity [25, 26]. *The ideal Bose gas (the weak-coupling limit) thus captures a good part of the important features of superfluidity (the finite-coupling case).* That the introduction of the interaction does not affect these features was established through the development of microscopic understanding of the superfluidity, in particular through works by Feynman, Penrose and Onsager [27, 28, 29, 21]. The validity of Feynman's approach was later confirmed quantitatively by direct Monte Carlo simulations [30, 31, 32, 33].

We will consider in this section the ideal Bose gas trapped in a harmonic potential in $d$ spatial dimensions [34], as the system closely resembles that of the O($N$) vector model studied in the previous section. There are $N$ harmonic oscillators denoted by $\vec{x}_1, \cdots, \vec{x}_N$, each of them having $d$ components. The Hamiltonian is

$$H = \sum_{c=1}^{N} \left( \frac{\vec{p}_c^2}{2m} + \frac{m\omega^2}{2} \vec{x}_c^2 \right). \tag{19}$$

Because the $N$ particles are indistinguishable bosons, invariance under permutations $S_N$ is imposed, which can equivalently be interpreted as gauging the $S_N$ symmetry. We note that the field $(\phi_1, \cdots, \phi_N)$ of the gauged O($N$) vector model discussed in the previous subsection is the counterpart of $(\vec{x}_1, \cdots, \vec{x}_N)$ in the present model. Both $x$'s and $\phi$'s belong to the fundamental representation of the gauge groups $S_N$ and O($N$), respectively. If we identify the fields $x$ and $\phi$, $S_N$ is naturally embedded into O($N$). In this sense, one may refer to the model considered in this section as '$S_N$ vector quantum mechanics'. This close similarity between the gauged O($N$) vector model and the system of identical bosons is what makes the O($N$) model particularly suited to connect the idea of confinement and BEC.

In the thermodynamic limit of the grand canonical ensemble, the number of particles in the excited states $M$ is given by

$$M = \int_0^\infty d\epsilon \frac{c_d \epsilon^{d-1}}{e^{\beta(\epsilon-\mu)} - 1}, \tag{20}$$

where $c_d = \frac{1}{(d-1)! \cdot \omega^d} = \frac{1}{\Gamma(d) \cdot \omega^d}$. The chemical potential $\mu$ has to satisfy $\mu \leq 0$. As a function of $\mu$, $M$ is monotonically increasing. The largest possible value is given at $\mu = 0$. Hence, if $M(\mu = 0) < N$, a BEC is formed; *a large fraction of particles, namely $N - M$ of them, are in the ground state.*

These states dominating the condensed phase can be written in a form that is analogous to eq. (1) and eq. (9). To this end, we introduce a set of basis vectors of the system (before imposing the $S_N$ gauge symmetry, *i. e.*, the complete symmetry under the exchanging of particles),

$$|\vec{n}_1, \vec{n}_2, \cdots, \vec{n}_N\rangle \equiv \prod_{i=1}^{d} \frac{\hat{a}_{i1}^{\dagger n_{i1}}}{\sqrt{n_{i1}!}} \frac{\hat{a}_{i2}^{\dagger n_{i2}}}{\sqrt{n_{i2}!}} \cdots \frac{\hat{a}_{iN}^{\dagger n_{iN}}}{\sqrt{n_{iN}!}} |0\rangle. \tag{21}$$

The state of each particle in the $d$-dimensional harmonic oscillator potential is specified by a $d$-dimensional interger-valued vector $\vec{n}$, where $n_i = 0, 1, \cdots$ with $i = 1, \cdots, d$. The $n_{i1}$

above specifies the state of particle 1, and so forth. By using this notation, the states in the condensed phase are,

$$P|\vec{n}_1, \cdots, \vec{n}_M, \vec{0}, \cdots, \vec{0}\rangle \tag{22}$$

Here $\hat{g}$ is the group element $g \in S_N$ represented as a unitary operator acting on the Hilbert space. and $\hat{P} = \frac{1}{N!}\sum_{g\in S_N} \hat{g}$ is the projection operator (the complete symmetrization operator). The un-symmetrized state $|\vec{n}_1, \cdots, \vec{n}_M, \vec{0}, \cdots, \vec{0}\rangle$ is analogous to $|E; \mathrm{SU}(M)\rangle$ in eq. (1) and $|E; \mathrm{O}(M)\rangle$ in eq. (9), shown pictorially in Fig. 1.

In the following, we will denote $M(\mu = 0)$ simply by $M$. By using $\int_0^\infty du \frac{u^{d-1}}{e^u - 1} = \zeta(d)\Gamma(d)$, we obtain

$$M = \frac{T^d \zeta(d)}{\omega^d}, \tag{23}$$

and hence the transition temperature $T_c$ is determined by,

$$N = \frac{T_c^d \zeta(d)}{\omega^d}. \tag{24}$$

Therefore we have

$$\frac{M}{N} = \left(\frac{T}{T_c}\right)^d \tag{25}$$

and

$$T_c = \left(\frac{N}{\zeta(d)}\right)^{1/d} \omega. \tag{26}$$

Curiously, these formulae are almost identical to the corresponding ones for the $\mathrm{O}(N)$ vector model, (17) and (18).

The energy below $T_c$ is

$$E = \int_0^\infty d\epsilon \frac{c_d \epsilon^d}{e^{\beta\epsilon} - 1} = c_d T^{d+1} \zeta(d+1)\Gamma(d+1). \tag{27}$$

We can easily see that, at $T \leq T_c$,

$$E(T = T_c(M)) = E_c(M). \tag{28}$$

The entropy satisfies a similar relation,

$$S(T = T_c(M)) = S_c(M). \tag{29}$$

Evidently, both energy and entropy are carried solely by excited modes. These relations are the counterpart of (16). In this way, *the thermodynamic properties of the system can be understood by the two-component (particles in the ground state and in the excited states)* .

Thus far, we observed a striking similarity between BEC in the system of $N$ identical bosons, and confinement both in the $\mathrm{O}(N)$ vector model and Yang-Mills theory. In the next section, we will describe the common mechanism that underlies BEC and confinement. Along the way, we will introduce the counterpart of (15) in BEC.

# 3 Common underlying mechanism

We now proceed to explain the common mechanism of color confinement and BEC, which is the origin of the remarkable similarity between these phenomena described in Sec. 2.

As a result of the constraint on the permutation symmetry, the system of bosons favors states where many particles are in the same state. This well-known enhancement effect, sometimes denoted as positive interference of the wave functions, is the essential mechanism responsible for BEC. Although a good part of the explanations that we provide below is well-known in the context of BEC, nonetheless, we decide to present it at a level of detail that exposes the connection to partial confinement in gauge theory.

For a system of $N$ indistinguishable bosons, permutation invariance can be incorporated by introducing a projection factor into the partition function as

$$Z = \sum_{g \in G} \text{Tr}\left(\hat{g} e^{-\beta \hat{H}}\right), \tag{30}$$

where $G = \text{S}_N$. Here, $\hat{g}$ is the group element $g \in G$ represented as a unitary operator acting on the Hilbert space. The inclusion of the projection factor allows for the trace to be taken over the full Hilbert space and as a complete orthonormal basis, we can use $|\vec{n}_1, \vec{n}_2, \cdots, \vec{n}_N\rangle$ defined by eq. (21).

As explained in Appendix A, the partition function can equivalently be obtained by summing the contributions from the permutation-invariant states, proportional to $\hat{P}|\vec{n}_1, \cdots, \vec{n}_N\rangle$, where $\hat{P} = \frac{1}{N!} \sum_{g \in \text{S}_N} \hat{g}$ is the projection operator. For generic excited states (in which no two particles occupy the same state) the sum over $g$ in (30) is used up for making the state completely symmetric. On the other hand, the ground state $|\vec{0}, \cdots, \vec{0}\rangle$ is genuinely symmetric, even before summing over $g$. This difference is the foundation of the enhancement effect. We can write the sum more explicitly as

$$
\begin{aligned}
Z &= \sum_{g \in \text{S}_N} \sum_{\vec{n}_1, \cdots, \vec{n}_N} \langle \vec{n}_1, \cdots, \vec{n}_N | \hat{g} e^{-\beta \hat{H}} | \vec{n}_1, \cdots, \vec{n}_N \rangle \\
&= \sum_{\vec{n}_1, \cdots, \vec{n}_N} e^{-\beta \left(E_{\vec{n}_1} + \cdots E_{\vec{n}_N}\right)} \sum_{g \in \text{S}_N} \langle \vec{n}_1, \cdots, \vec{n}_N | \hat{g} | \vec{n}_1, \cdots, \vec{n}_N \rangle \\
&= \sum_{\vec{n}_1, \cdots, \vec{n}_N} e^{-\beta \left(E_{\vec{n}_1} + \cdots E_{\vec{n}_N}\right)} \sum_{g \in \text{S}_N} \langle \vec{n}_1, \cdots, \vec{n}_N | \vec{n}_{g(1)}, \cdots, \vec{n}_{g(N)} \rangle.
\end{aligned} \tag{31}
$$

If all $N$ particles are in different states, only $g = \mathbf{1}$ gives rise to a nonzero contribution. On the other hand, if all $N$ particles are in the same state, all $g$'s return the same nonzero contribution, leading to an enhancement factor of $N!$ compared to the case where we do not impose the gauge singlet constraint (or equivalently the classical Boltzmann statistics).

One can also think of this enhancement as a consequence of redundancy in gauge theories: configurations connected by a gauge transformation are identified. Equivalently, when the system of $N$ identical bosons is regarded as an '$\text{S}_N$ gauge theory', 'states' connected by

a gauge transformation (i.e. a permutation) — $|\vec{n}_1, \cdots, \vec{n}_N\rangle$ and $|\vec{n}_{g(1)}, \cdots, \vec{n}_{g(N)}\rangle$ — should be identified. Consider for example $|\vec{n}, \vec{0}, \cdots, \vec{0}\rangle$, $|\vec{0}, \vec{n}, \vec{0}, \cdots, \vec{0}\rangle$, ..., $|\vec{0}, \cdots, \vec{0}, \vec{n}\rangle$. A priori, these are $N$ different states. Once the $S_N$ symmetry is gauged, they are identified and their statistical weight is reduced to precisely match that of the ground state. Explicitly, one sees that the aforementioned enhancement factors, here $(N-1)!$, combine with the degeneracy factor $N$ to yield an overall coefficient of $N!$. In comparison, the ground state is unique but is enhanced by a factor of $N!$. For generic excited states, there is an $N!$-fold over-counting which is compensated by the absence of the enhancement factor. [14] Thus all gauge invariant states contribute with equal weight. As a result, the relative importance of configurations where many particles occupy the same state is significantly increased, compared to a system of distinguishable particles where the gauge singlet constraint is not imposed.

This mechanism directly carries over to ordinary gauge theory, where the gauge-singlet constraint (or equivalently, the Gauss law constraint) is introduced in the same fashion. Now, $g$ in (30) is an element of the gauge group, e.g. O($N$) or SU($N$), and the sum is replaced by the invariant integral over the gauge group. The group element $g$ now coincides with the Polyakov loop.[15] From this prescription, we see that an enhancement mechanism that is essentially equivalent to the one in BEC also applies to the gauge theory. Namely, if all degrees of freedom are in their ground states (the fully confined state), the integral over the gauge group gives a larger factor compared to states where degrees of freedom are in different excited states (the deconfined state). This argument applies not just to fields in the fundamental representation, but also to those in the adjoint representation, such as gluons. Thus, the confined state, rather than the deconfined state, is favored as a result of the gauge-singlet constraint. Note that this argument readily generalizes to the interacting theory, as long as *the confined sector provides positive interference.*

Note that this mechanism can work for QFT in any spacetime dimensions. The important point is that local gauge symmetry leads to an enhancement factor at each spatial point. To make the story well-defined by introducing a proper regularization, we can use the lattice Hamiltonian, e.g. the Kogut-Susskind formulation. At finite lattice spacing and finite lattice points, it is just a 'matrix model' consisting of many matrices (link variables and site variables). Therefore, strictly speaking, $\hat{g}$ is $\hat{g} = \otimes_{\vec{x}} \hat{g}_{\vec{x}}$, where $\hat{g}_{\vec{x}}$ is the group element associated with a point $\vec{x}$. Each $\hat{g}_{\vec{x}}$ can be regarded as the Polyakov loop at point $\vec{x}$. See Appendix C how the symmetrization over the local gauge transformation leads to the gauge singlets.

That large-$N$ Yang-Mills theory deconfines at higher temperature is usually understood

---

[14]More generally, if $M$ particles are excited to different excitation levels while $N - M$ particles are in the ground state, classically there are $\frac{N!}{(N-M)!}$ different states, and the enhancement factor is $(N-M)!$. Hence the weight in the partition function does not depend on $M$.

[15]A simple way to understand this is to consider lattice regularization and take the $A_0 = 0$ gauge. The unitary link variable along the time direction $U_t$ connecting the Euclidean time $t$ and $t + a$, where $a$ is the lattice spacing, transforms as $U_t \rightarrow \Omega_t U_t \Omega_{t+a}^{-1}$. We can use this to set all links to unity, except for the one at $t = 0$ which is by definition the Polyakov loop.

as a consequence of the Hagedorn growth of the density of states, $\Omega(E) \sim e^{\frac{E}{T_H}}$ [6, 7] at $E \lesssim N^2$, where $T_H$ is the Hagedorn temperature [35]. This particular growth rate with respect to the energy allows for a dramatic growth of the energy and entropy as functions of temperature, at $T = T_H$, from order $N^0$ to order $N^2$. At infinite $N$ this is the well-known Hagedorn growth that is obtained by counting the number of singlet states, using the chromoelectric string picture. If the singlet constraint were not imposed, the density of states is always $\mathcal{O}(N^2)$ and no Hagedorn-like growth can be observed. The mechanism explained in the previous paragraph gives a complimentary understanding of the confinement/deconfinement transition. Either way one observes the effect of the singlet constraint but from different angles.

This relationship between confinement and BEC gives us a better understanding of why partial confinement occurs as depicted in Fig. 1. Consider, for example, the possibility that the deconfined sector is given by two diagonal blocks whose sizes equal $M_1, M_2$ with $M_1^2 + M_2^2 \approx M^2$, while the remaining matrix elements are confined. Naively, such states would have the same entropy as those shown in Fig. 1, because the numbers of excited matrix entries are the same. We see now that this type of partial confinement pattern is ruled out because the volume of the group SU$(N - M_1 - M_2)$ is much smaller than SU$(N - M)$, and hence the enhancement effect is much smaller; therefore these states cannot dominate thermodynamics.

# 4 Polyakov loop and off-diagonal long range order

In the previous sections, we have pointed out that BEC and partial confinement in large $N$ gauge theories share the essential features listed in the introduction, based on the discussion in the weak-coupling limit. In this section, we will consider how our argument can be extended to interacting theories. For BEC, in the presence of inter-particle interactions, Hamiltonian eigenstates are of course no longer given by symmetrized products of individual particle states. As a consequence, it is not immediately clear how to define, for interacting theory, 'the number of particles in their ground states' which characterizes the condensed phase for the ideal gas. Penrose and Onsager [21] proposed a criterion valid for interacting theories, later referred to as 'Off-Diagonal Long Range Order' (ODLRO) [22], which utilizes a natural extension of the concept of 'the number of particles in their ground states'. For gauge theories, on the other hand, the distribution of Polyakov loop phases, as explained in the previous section, provides a good criterion for partial confinement, applicable also to the interacting case [14]. We will now show that ODLRO in BEC and the Polyakov loop in gauge theories are closely related. Along the way, we will demonstrate that one can define ODLRO for gauge theories, and a Polyakov loop for BEC.

## 4.1 Off-diagonal long range order

We begin by recalling the definition of ODLRO for $N$ identical bosons. Denoting the density matrix of the $N$-particle system by $\hat{\rho}$, the one-particle density matrix is defined by tracing out $N - 1$ particles, $\hat{\rho}_1 = N \cdot \mathrm{Tr}_{2,3,\cdots,N}\hat{\rho}$. It can be conveniently written via its spectral decomposition,

$$\hat{\rho}_1 = n_{\max}|\Psi\rangle\langle\Psi| + \sum_i n_i|\Psi_i\rangle\langle\Psi_i|, \tag{32}$$

where $n_{\max}$ is the largest eigenvalue and $|\Psi\rangle$ is the corresponding eigenvector. The eigenvectors $|\Psi\rangle$ and $|\Psi\rangle_i$ are normalized to be unit norm. When $n_{\max}$ is of order $N$, the system contains a BEC, and is characterized by ODLRO. For a BEC of non-interacting bosons, $|\Psi\rangle$ is the one-particle ground state, and we have $n_{\max} = N - M$, i. e. the number of particles in the ground state.

In the usual thermodynamic limit with fixed particle density, $V \sim \omega^{-d} \sim N$, the reduced density matrix $\langle x|\hat{\rho}_1|y\rangle$ is non-vanishing at long distance if $n_{\max}$ is of order $N$. The order is associated with the off-diagonal matrix elements in the coordinate representation; this is the origin of the name of ODLRO.

## 4.2 Polyakov loop for identical bosons

Let us start with the partition function (30). Again, a convenient basis is (21). Let $M_{\vec{n}}$ ($\sum_{\vec{n}} M_{\vec{n}} = N$) be the number of particles in the state specified by $\vec{n}_i = \vec{n}$. A permutation $\{g \in \prod_{\vec{n}} \mathrm{S}_{M_{\vec{n}}}\}$ leaves the corresponding state invariant and gives rise to a nonzero contribution to (30). As we have mentioned, this $g$ is the counterpart of the Polyakov loop in gauge theory. The distribution of the phases of this 'Polyakov loop' can be obtained by calculating the average eigenvalue distribution of $\{g = \{g_{\vec{n}}\} \in \prod_{\vec{n}} \mathrm{S}_{M_{\vec{n}}}\}$. At large $N$, we can use the typical values of $M_{\vec{n}}$ realized in the BEC.

As $M_{\vec{0}} \sim N \to \infty$ (i.e. as the BEC is formed), the average eigenvalue distribution of $g_{\vec{0}} \in \mathrm{S}_{M_{\vec{0}}}$ becomes uniform. To see this, let us note that when $g$ is a cyclic permutation of $k$ elements, the eigenvalues of $g$ are $e^{2\pi i l/k}$, $l = 0, 1, \cdots, k - 1$. When $k \to \infty$, the phases are distributed uniformly and continuously between $-\pi$ and $+\pi$. Any $g_{\vec{0}} \in \mathrm{S}_{M_{\vec{0}}}$ can be written as a product of cyclic permutations of different sets of elements, and as $M_{\vec{0}} \to \infty$, infinitely long cyclic permutations become dominant.[16] Therefore, $g_{\vec{0}} \in \mathrm{S}_{M_{\vec{0}}}$ leads to a uniform distribution. This is the counterpart of $\frac{1}{2\pi}\left(1 - \frac{M}{N}\right)$ in partial deconfinement. Thus we have shown that the particles in the ground state (which is measured by ODLRO) contribute to the constant offset of the Polyakov loop.

In order to complete the proof of equivalence of the constant offset to the number of particles in the ground state as measured by ODLRO, it remains to be shown that the

---

[16]Importance of the dominance of long cyclic permutation in understanding BEC for interacting bosons is first pointed out by Feynman in his microscopic theory of superfluidity of $^4$He [27]. The presence of ODLRO when the long cyclic permutation dominates is shown in [21].

particles in excited modes do not contribute to the constant offset. This is somewhat intricate due to the discreteness of the permutation group $S_N$ and we defer the reader to Appendix B for a detailed proof. With this in hand, we can directly read off the number of condensed particles from the Polyakov loop. Such a formulation, based on the Polyakov loop, has the advantage that *one can infer the existence of positive interference from the nonzero constant offset, regardless of the details of the interaction.* Even at strong coupling, the same quantity characterizes the number of degrees of freedom in the BEC sector.[17]

## 4.3   Polyakov loop in gauge theory and ODLRO

In the case of a gauge theory, the partition function is given by (30) with $G$ now denoting the gauge group, e.g. $O(N)$ or $SU(N)$. As mentioned before, $g$ corresponds to the Polyakov loop. The ground state is responsible for the constant distribution, because a generic element in $O(N)$ or $SU(N)$ gives a uniform distribution at large $N$. Hence we can count the number of degrees of freedom in the confined sector.[18] This argument applies to any large-$N$ gauge theory regardless of the details of the field content; that the distribution of the Polyakov loop phases becomes uniform in the confined phase demonstrates the strong positive interference. The constant offset (the minimum of the distribution) is related to the size of the deconfined sector via

$$\text{The constant offset} = 1 - \frac{M}{N}. \tag{33}$$

This is the order parameter[19] of the partial confinement. The lattice simulations of the bosonic matrix model [36, 37] provide a concrete example at strong coupling.

Naturally, we can also define a counterpart of ODLRO for gauge theories, via a reduced 'one-color' density matrix. For example we can keep only the zero-mode of one of the color degrees of freedom (say the first component of the matter field in the fundamental representation, or $(1, 1)$-component of the adjoint field) and trace out all other degrees of freedom. The existence of the confined phase can then be read off from the largest eigenvalue of the reduced density matrix[20]. If we normalize the largest eigenvalue of the reduced density matrix in such a way that it equals unity for the fully confined (condensed) phase, then it corresponds to the constant offset of the Polyakov loop. We note that "the

---

[17]Here an implicit assumption is that excited modes do not contribute to the constant offset, which may fail when many light degrees of freedom exist.

[18]This was known in several weakly-coupled theories via explicit analytic calculation [14, 11, 15], but there was no concrete justification.

[19]Polyakov loop is often used as an order parameter to detect the spontaneous breaking of the center symmetry. Here we are using the Polyakov loop as the order parameter in a different way. Not only it applies to theories without the center symmetry, it is more precise in the sense that it can distinguish three phases: completely-confined, partially-confined, and completely deconfined.

[20]This can be done in a gauge invariant manner, in the same way as the one-particle reduced density matrix is permutation invariant in the case of identical bosons. The density matrix itself $\rho = \sum |\Psi_i\rangle e^{-\beta E_i} \langle \Psi_i|$ is gauge invariant, $|\Psi_i\rangle$ satisfying the Gauss law constraint. Because of this there is no ambiguity from the choice of gauge when defining the eigenvalues of the reduced density matrix.

long range order" in this context is longe range not in the spacetime but in the 'emergent space' described by the values of the field.

We expect that the large positive interference responsible for the constant offset survives when the interaction is turned on adiabatically, just like ODLRO does. Both order parameters (the eigenvalue in ODLRO, and the constant offset of the Polyakov loop) are tied to the gauge symmetry as explained in Sec. 3. Because of this, we expect that for any value of the coupling constant the two transitition points, namely, from completely-confined to partially-confined phase, and from partially-confined to the completey-deconfined phase, should be captured by the conditions that the order parameters be equal to 0 and 1, respectively.

# 5  Discussions

In this paper, we pointed out that two important phenomena, BEC and (partial) confinement, can be understood in a unified way. We expect that, because of this new connection, computational tools, and perhaps more importantly intuition developed for one of them can now enrich the understanding of the other. For example, in superfluidity, transport properties are well understood in terms of a two fluid model corresponding to condensed and excited states; can we obtain a similar understanding for the transport properties in a partially confined phase?

We have focused on model-independent features, such as the mechanism behind the phenomenon and its essential characterization. Confinement (condensation) occurs because a large fraction of the degrees of freedom fall into the ground state. This phase is favored because of the large interference effect originating in the gauge symmetry. More detailed features, such as the precise structure of the phase diagram (including the existence of a completely condensed phase) and the order of the phase transition, depend on model specifics. [21]

Our strategy has been to understand confinement by an adiabatic continuation of the weak-coupling (small volume) picture. Whether this picture remains relevant at strong coupling (large volume) depends on the dynamics of the model and in particular relies on the absence of a phase boundary that obstructs the interpolation between strong- and weak-coupling regions. However, there are indications that such an obstruction is absent for various theories important for gauge/gravity duality, most notably 4d $\mathcal{N} = 4$ super Yang-Mills, although thus far there exists no direct proof.[22] Whether the strong and weak-

---

[21] A classic example of this type of model dependence is the difference between the superfluidity of $^4$He and the condensation of an ideal Bose gas. For the ideal gas the transition is of third order, whereas the $\lambda$-transition of $^4$He is of second order. The ideal bose gas is completely condensed at $T = 0$, whereas $^4$He is not. Nevertheless, they share common characterization (such as ODLRO) and mechanism, and the analogy to BEC of the ideal Bose gas was an important step to understand superfluidity.

[22] On the other hand, QCD with too many flavors is conformal at infrared, which suggests the strong dynamics spoils the weak-coupling picture in this case.

coupling regimes are smoothly connected for a given model is a question which can be tested by lattice Monte Carlo simulations. For the D0-brane quantum mechanics and its plane wave deformation, extensive numerical studies have been performed, starting from Refs. [38, 39], which support the absence of the obstruction.

The analogy to BEC also provided new insight on order parameters which should be useful to interpolate between the weak and the strong-coupling regimes. We showed that the constant offset of the distribution of the Polyakov loop phases corresponds to ODLRO, and is tied to the structure of the gauge symmetry associated with the condensed phase. This gives in particular a characterization of partial confinement which is valid even at nonzero coupling. The constant offset $\frac{N-M}{N}$ is the order parameter that encodes the size of the deconfined sector: an SU($M$)-subsector of SU($N$)-theory is deconfined.

## Implication for gauge theories; Connection to QCD?

In BEC for interacting bosons, 'the number of particles in the ground state' as defined by ODLRO is less than $N$ even at zero temperature, which is in marked contrast with the ideal Bose gas where for $T = 0$ all particles are in the ground state. An intriguing possibility is that a similar phenomenon may occur for some gauge theories: for these theories there may not be complete confinement even for $T = 0$.

It is interesting to study the connection of our understanding of confinement as BEC to the more traditional pictures of dynamical confinement (e.g. based on the linear potential between quarks). It might be possible to achieve this through the idea of magnetic monopole condensation [40, 41, 42, 43] which is a promising scenario for dynamical confinement. In some versions of this scenario, singularities plays an important role that occurs when the nature of the degrees of freedom associated with the monopole changes (e. g. when the monopole becomes massless) [42, 43]. What happens at these singularities resembles the enhancement effect of confined states in our scenario because of the large interference effect. Namely, in the partition function (30), while the confined sector is genuinely SU($N - M$)-symmetric, the deconfined sector is SU($M$)-symmetric only due to symmetrization. In other words, the confined sector is statistically enhanced (positive interference) while the deconfined sector is not (see also Appendix A).

Given that theories at small volume and large $N$ are often quantitatively close to those at large volume and moderate $N$ [44, 45, 46, 47, 48], it seems imaginable to also interpret confinement at finite $N$ as BEC. For example, in the Twisted Eguchi-Kawai reduction [45], large-$N$ theory at small volume behaves similar to the finite-$N$ theory at volume $V \sim N^2$. Closely related phenomena have been studied extensively in QCD-like theories with adjoint fermion [49] or certain deformation terms [50]. Such theories would provide us with analytically controllable setups. Recall that for indistinguishable bosons in a harmonic trap, the thermodynamic limit $V \sim \omega^{-d} \sim N$ is typically taken with fixed particle density. In this limit, interference effects contribute to the free energy with a relative factor $\log(N!) \sim V(\log V - 1)$. In gauge theory, because the gauge group can act locally, even when $N$ is fixed there is a similar factor $\sim V \log V_G$, where $V_G$ is the volume of gauge group $G$. One

should be able to understand confinement for finite $N$ gauge theories as the result of a mechanism similar to that discussed in Sec. 3, because of this large enhancement factor.

Finally, it will be an important step to investigate possible experimental signals in colliders that could indicate whether confinement in actual QCD bears any resemblance with BEC.

## Condensation of D-branes?

D-branes play essential roles in string theory. As is well-known, their low-energy effective theory is a certain Yang-Mills theory coupled to adjoint matter fields with a U($N$) gauge group [51]. Diagonal elements of the adjoint scalar fields corresponds to the location of D-branes. This U($N$) group contains $S_N$ subgroup which permutes D-branes. In this sense, U($N$) gauge symmetry can be interpreted as generalization of $S_N$ permutation symmetry. Consider now the system of D-branes at very low temperature such that the typical distance between them is smaller than their thermal de Broglie wavelength. In this regime, it is natural to expect that the D-branes would undergo a quantum statistical transition, analogous to BEC.[23] The similarity of partial confinement to BEC advocated in this paper makes it plausible that partial confinement should be crucial in the understanding of this quantum condensation of D-branes.

## Holographic emergent space?

In the standard interpretation, the completely deconfined and confined phases correspond to the AdS vacuum and a black hole, respectively [4]. A natural candidate for a dual gravity interpretation of partially deconfined and confined sectors are the small black hole and its exterior [12, 14, 11]. According to the analogy to BEC, the small black hole would correspond to a droplet of normal fluid within superfluid. The Hawking radiation then will be analogous to the dissipation of this droplet.

In the case of four-dimensional $\mathcal{N} = 4$ super Yang-Mills, the six scalar fields can condense. Such a BEC is effectively six-dimensional at each point in 3d space, thus leading to nine-dimensional space. One may speculate that gravity can be understood as collective excitations analogous to phonons in superfluid helium. Such an interpretation would provide us with a natural generalization of the philosophy of the Matrix Model of M-theory (BFSS) [52] — physical objects are realized as sub-matrices — to gauge/gravity duality à la Maldacena. Note also that partial deconfinement is naturally connected to Higgsing; when a deconfined block is far separated (in the sense of eigenvalues), Higgsing is a better descrip-

---

[23]Although D-branes are so-called superparticles that can be bosonic or fermionic depending on the excitation of their internal degrees of freedom, the bosonic degrees of freedom will dominate for low temperature physics we are interested in. This is because states associated with the fermionic degrees of freedom inevitably have much higher energy than their bosonic counterparts, since they live on the Fermi surface due to the Pauli principle.

tion because the off-diagonal elements become heavy and decouple from the dynamics.[24] When the partially-deconfined sector represents a D-brane probe, it should be described by the Dirac-Born-Infeld action on $AdS_5 \times S^5$, as proposed in Ref. [54]. Furthermore note that the color degrees of freedom in the confined sector can be entangled and naturally lead to a picture for emergent space [55, 11] along the lines of Refs. [56, 57]. When colors are identified with qubits, 'it from qubit' naturally meets the good old idea of 'everything from matrices'. One may hope that the intuition gained by connecting BEC and confinement will be a useful guide towards understanding the nature of the building blocks of emergent spacetime.

# Acknowledgement

We would like to thank Yoichi Kazama for critical comments and useful discussions. We also thank Ofer Aharony, Sinya Aoki, Andy O'Bannon, Brandon Robinson, Andreas Schmitt, Kostas Skenderis, Bo Sundborg, and Naoki Yamamoto for stimulating discussions and Etsuko Itou, Antal Jevicki, Paul Romatschke, and Stephen Shenker for carefully reading the draft. MH was supported by the STFC Ernest Rutherford Grant ST/R003599/1 and JSPS KAKENHI Grants17K1428. NW acknowledges support by FNU grant number DFF-6108-00340.

# A    Another look at positive interference

In order to understand positive interference further, let us see how the partition function (31) for $N$ free bosons is obtained by summing the contribution of permutation-invariant states. By using the projection operator $\hat{P} = \frac{1}{N!} \sum_{g \in S_N} \hat{g}$, we can write the invariant states as

$$c_{\vec{n}_1, \cdots, \vec{n}_N}^{-1} \times \hat{P} |\vec{n}_1, \cdots, \vec{n}_N\rangle. \tag{34}$$

where $c_{\vec{n}_1, \cdots, \vec{n}_N}$ ensures unit normalization. For example, when $N = 2$, $c_{\vec{n}_1, \vec{n}_2} = 1$ for $\vec{n}_1 = \vec{n}_2$ and $c_{\vec{n}_1, \vec{n}_2} = \frac{1}{\sqrt{2}}$ for $\vec{n}_1 \neq \vec{n}_2$. In general, if there are $l$ different one-particle states with degeneracies $N_1, \cdots, N_l$ $(N_1 + \cdots + N_l = N, N_i \geq 1, l < N)$,

$$c_{\vec{n}_1, \cdots, \vec{n}_N} = \sqrt{\frac{\prod_{i=1}^{l} N_i!}{N!}}. \tag{35}$$

Note that this factor $\prod_{i=1}^{l} N_i!$ is related to positive interference.

When we calculate the partition function, if we took the sum with respect to any $\vec{n}_1, \cdots, \vec{n}_N$, we would be counting the same state multiple times, with the over-counting

---

[24]It is well-known that, in presence of scalar matter fields, the confinement phase is smoothly connected to the Higgs phase [53].

factor $\frac{N!}{\prod_{i=1}^{l} N_i!}$. By compensating this factor, we obtain

$$
\begin{aligned}
\sum_{\vec{n}_1,\cdots,\vec{n}_N} & \left(\frac{N!}{\prod_{i=1}^{l} N_i!}\right)^{-1} \cdot c_{\vec{n}_1,\cdots,\vec{n}_N}^{-2} \cdot \langle \vec{n}_1,\cdots,\vec{n}_N|\hat{P}e^{-\beta\hat{H}}\hat{P}|\vec{n}_1,\cdots,\vec{n}_N\rangle \\
&= \sum_{\vec{n}_1,\cdots,\vec{n}_N} \langle \vec{n}_1,\cdots,\vec{n}_N|\hat{P}e^{-\beta\hat{H}}\hat{P}|\vec{n}_1,\cdots,\vec{n}_N\rangle \\
&= \sum_{\vec{n}_1,\cdots,\vec{n}_N} \langle \vec{n}_1,\cdots,\vec{n}_N|\hat{P}e^{-\beta\hat{H}}|\vec{n}_1,\cdots,\vec{n}_N\rangle \\
&= \frac{1}{N!}\sum_{\vec{n}_1,\cdots,\vec{n}_N}\sum_{g\in\mathrm{S}_N} \langle \vec{n}_1,\cdots,\vec{n}_N|\hat{g}e^{-\beta\hat{H}}|\vec{n}_1,\cdots,\vec{n}_N\rangle.
\end{aligned}
\tag{36}
$$

This is eq. (31), up to the overall factor $(N!)^{-1}$. For gauge theory, the symmetrization defined by eq. (1) does exactly the same job: the symmetrized state in eq. (1) is the counterpart of (34). In Sec. 2.1, we started with the SU($M$)×SU($N-M$)-invariant state $|E; \mathrm{SU}(M)\rangle$. That approach is advantageous for the computation of the entropy. However, one can start with a state without imposing the Gauss law constraint associated with SU($M$); The symmetrization (1) will assure the SU($M$)-invariance in the deconfined sector. The deconfined sector is SU($M$)-invariant due to the symmetrization, in the same way that the excited sector of the system of identical bosons is $\mathrm{S}_M$-invariant. In contrast, the confined sector is 'genuinely' gauge-invariant, even without symmetrization, and hence, the enhancement factor, which is the volume of SU($N-M$), appears.

# B  More on ODLRO and Polyakov loop

In this appendix, we prove the equivalence of ODLRO and the criteria based on the Polyakov loop for the ideal gas in a harmonic oscillator potential well by showing that excited modes do not contribute to the constant offset of the Polyakov loop.

For this purpose, it is convenient to express the partition function in the following form. Suppose that a given element $g \in \mathrm{S}_N$ is a product of cyclic permutations with length $l_1, l_2, \cdots$. Then,

$$
\mathrm{Tr}\left(\hat{g}e^{-\beta\hat{H}}\right) = \prod_i \left(\frac{1}{1-e^{-l_i\beta\omega}}\right)^d.
\tag{37}
$$

Let $N_l$ be the number of cyclic permutations with length $l$. Then the partition function (30) is written as

$$
Z = \sum_{\{N_l\}} \frac{N!}{\prod_l \left(l^{N_l} \cdot N_l!\right)} \left(\frac{1}{1-e^{-l\beta\omega}}\right)^{dN_l}.
\tag{38}
$$

Here the sum is taken over all possible $\{N_l\}$ satisfying $\sum_l l N_l = N$. We introduce a Lagrange multiplier (chemical potential) $\mu$ to enforce this constraint and minimize the free energy to obtain

$$N_l = \frac{e^{-\mu l}}{l(1 - e^{-l\beta\omega})^d}, \qquad \sum_{l=1}^{N} l N_l = N. \tag{39}$$

In the large $N$ limit, the Polyakov loop $g$ specified with these $N_l$'s dominates.

The eigenvalue distribution of $\hat{g}$ is determined by $N_l$ as explained in the main text. Namely, for each cyclic permutation with length $l$, there are $l$ numbers of eigenvalues, $e^{2\pi i k/l}$, $k = 0, 1, \cdots, l - 1$. The total number of the eigenvalues is, of course, $\sum l N_l = N$. Our task is to understand the distribution function $\rho(\theta)$ of the phases of the eigenvalues determined by (39) in the large $N$ limit. We choose $\theta$ to be in the range $0 \le \theta < 2\pi$ for convenience and normalize $\rho(\theta)$ by $\int \rho d\theta = 1$.

We will first consider the case $T = T_c$. We denote $\rho(\theta)$ in this critical case as $\rho_c(\theta)$. The statement we wish to prove first is that the constant offset, i.e. the minumum of $\rho_c(\theta)$, vanishes. Note that $\rho_c(\theta) \ge 0$ by definition. Our strategy is as follows: we will write the distribution $\rho_c(\theta)$ as a sum of two terms, namely, contributions from $l < \Lambda$ and $l \ge \Lambda$.

$$\rho_c(\theta) = \rho_{c,l<\Lambda}(\theta) + \rho_{c,l\ge\Lambda}(\theta) \tag{40}$$

where $\Lambda$ is a large 'cutoff'. By definition $\rho_{c,l<\Lambda}(\theta) \ge 0$ and $\rho_{c,l\ge\Lambda}(\theta) \ge 0$. One may imagine that we are evaluating $\rho_c(\theta)$ as $\rho_{c,l<\Lambda}(\theta)$ up to a certain precision, or equivalently a finite resolution in $\theta$, of $\delta\theta \sim \frac{2\pi}{\Lambda}$. The larger the value of $\Lambda$ the more precise our evaluation of $\rho_c(\theta)$ will be. We will show that the minimum of $\rho_{c,l<\Lambda}(\theta)$ vanishes for any finite $\Lambda$, and $\rho_{c,l\ge\Lambda}(\theta)$ (and hence its constant offset) can be made arbitrarily small by choosing $\Lambda$ to be sufficiently large but of order $N^0$.

At $T = T_c$, since $\mu = 0$ and $\beta\omega = \left(\frac{\zeta(d)}{N}\right)^{\frac{1}{d}}$, the formula (39) yields

$$\lim_{N\to\infty} \frac{\sum_{l<\Lambda} l N_l}{N} = 1 - \sum_{l=\Lambda}^{\infty} \frac{1}{\zeta(d)l^d}, \tag{41}$$

for large $\Lambda$. It is essential that for $d > 1$, which is the condition for BEC to occur, the second term converges and is of order $O(\Lambda^{-(d-1)})$. For $\Lambda \sim N^0$, $\rho_{l<\Lambda}(\theta)$ is a sum of finite (i. e. of order $N^0$) number of delta functions (located at $2\pi k/l$, $k = 0, \cdots, l - 1$ where $l < \Lambda$). The minimum of $\rho_{c,l<\Lambda}(\theta)$ is zero, for any finite value of $\Lambda$. On the other hand $\rho_{c,l\ge\Lambda}(\theta)$ may approach a continuous function in the large $N$ limit. In particular $\rho_{c,l\ge\Lambda}(\theta)$ could have contributed to a constant offset for $N \to \infty$. However, since

$$\int \rho_{c,l\ge\Lambda}(\theta)d\theta = \sum_{l\ge\Lambda} \frac{l N_l}{N} \approx \sum_{l\ge\Lambda} \frac{1}{\zeta(d)l^d} = O(\Lambda^{-(d-1)}) \tag{42}$$

for large $N$, the function $\rho_{c,l\geq\Lambda}(\theta)$ itself can be made arbitrarily small by choosing sufficiently large (but of order $N^0$) $\Lambda$. Thus we have shown $\rho_c(\theta)$ have a vanishing constant offset in the large $N$ limit.

For $T \leq T_c(N)$, the BEC is formed. The statistical distribution of the excited particles (and therefore the contribution to $\rho(\theta)$ from the excited states) is identical to that for the system with $N = M$ if we fix $M$ by $T = T_c(M)$. The ground state contributes a constant term, $\frac{1}{2\pi}\left(1 - \frac{M}{N}\right)$, as explained in the main text. Therefore, we obtain

$$\rho(\theta) = \frac{1}{2\pi}\left(1 - \frac{M}{N}\right) + \frac{M}{N}\rho_{c,l<\Lambda}(\theta) + O(\Lambda^{-(d-1)}), \tag{43}$$

which may be considered as the counterpart of (15) in partial deconfinement. Again one can choose $\Lambda$ to be sufficiently large (but of order $N^0$) such that the last term is negligible. The formula shows that the contribution of the constant offset is solely from the ground state, which completes the proof.

# C More on the gauge-invariant states via the symmetrization

In this appendix we consider a gauge-invariant operator constructed using the Wilson line,

$$\sum_{i,j=1}^{N} \hat{q}_i(x)^\dagger \hat{W}_{ij}(x,y)\hat{q}_j(y) \tag{44}$$

where $\hat{q}$ represents a quark field and $\hat{W}(x,y)$ is a Wilson line connecting points $x$ and $y$. We show that this is obtained via the symmetrization over the gauge symmery. Specifically, let us see how this gauge-invariant combination is obtained from $\hat{q}_I(x)^\dagger \hat{W}_{IJ}(x,y)\hat{q}_J(y)$, where in the latter the sums over $I$ and $J$ are not taken. Firstly we symmetrize over the U($N$) symmetry at point $x$. The SU($N$) transformation is given by

$$\hat{q}_I(x)^\dagger \rightarrow \sum_{i=1}^{N} \hat{q}_i(x)^\dagger U_{iI}^\dagger(x), \tag{45}$$

$$\hat{W}_{IJ}(x,y) \rightarrow \sum_{i'=1}^{N} U_{Ii'}(x)\hat{W}_{i'J}(x,y), \tag{46}$$

and

$$\hat{q}_I(x)^\dagger \hat{W}_{IJ}(x,y)\hat{q}_J(y) \rightarrow \sum_{i,i'} \hat{q}_i(x)^\dagger U_{iI}^\dagger(x)U_{Ii'}(x)\hat{W}_{i'J}(x,y)\hat{q}_J(y). \tag{47}$$

After averaging over the Haar measure, $U_{iI}^\dagger(x)U_{Ii'}(x)$ is replaced with $\delta_{ii'}$. Hence the symmetrization over $SU(N)$ at point $x$ leads to

$$\hat{q}_I(x)^\dagger \hat{W}_{IJ}(x,y)\hat{q}_J(y) \to \sum_{i=1}^{N} \hat{q}_i(x)^\dagger \hat{W}_{iJ}(x,y)\hat{q}_J(y). \tag{48}$$

We can perform the symmetrization over $SU(N)$ at point $y$ as well, and obtain:

$$\sum_{i=1}^{N} \hat{q}_i(x)^\dagger \hat{W}_{iJ}(x,y)\hat{q}_J(y) \to \sum_{i=1}^{N}\sum_{j=1}^{N} \hat{q}_i(x)^\dagger \hat{W}_{ij}(x,y)\hat{q}_j(y). \tag{49}$$

In the same manner, various gauge-invariant operators, both local and nonlocal, are obtained via the symmetrization.

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
