# Peer review of "Color Confinement and Bose-Einstein Condensation"

_SciPost Physics_

## Round 2 · Referee Report · Anonymous · 2020-6-2

Strengths

1. The paper points out a strong similarity between the physics of Bose Einstein condensation and the confinement/deconfinement phase transition in large N gauge theories. The connection between permutation symmetry and gauge symmetry is provocative (and new at least to the referee).
2. Evidence is presented of the correctness of this conjecture by analysis of models at weak coupling by formulating them on small spheres.
3. The analogy to BEC suggest that methods used for the latter may be useful for understanding confinement in gauge theories.

Weaknesses

1. The models that can be treated this way require that the physics at weak coupling be continuously connected to strong coupling. While this may be true for certain supersymmetric theories it may not be true for other more physically relevant theories.
2. This analysis doesn't really lead to any new results that have not already been obtained using other methods. So the main point of the paper seems to be that it offers a new conceptual angle.

Report

I think the paper offers some new and worthwhile observations and conclusions that are most likely relevant to the case of theories
exhibiting gauge/gravity duality. I would have liked to see more discussion of eq. 1 - what is $\mathcal U$ that appears there ? I understand that integrating over all gauge transformations produces a singlet but there seems to be more content than that in this eqn.
Can one extend this analysis to finite (small) N by exploiting large
N volume independence ? Eguchi-Kawai reduction should hold for
many of the theories that are potentially being considered eg N=4 SYM,
models with fermions in the adjoint representation etc.
It would be nice for the authors to also include a brief discussion of these issues.

Requested changes

see above.

  • validity: high
  • significance: good
  • originality: high
  • clarity: good
  • formatting: excellent
  • grammar: excellent

Author Nico Wintergerst on 2020-07-20
(in reply to Report 1 on 2020-06-02)
Category:
remark
answer to question
pointer to related literature

Let us begin by thanking the referee for treating our submission with care and the considered analyses of its relative merits. In the following, we respond to the specific points raised by Referee 1 in their report, some of which have additionally been addressed in the revised version that is submitted in parallel. We have also taken this opportunity to correct typographical errors in appendix A.

[Strengths]

  1. The paper points out a strong similarity between the physics of Bose Einstein condensation and the confinement/deconfinement phase transition in large N gauge theories. The connection between permutation symmetry and gauge symmetry is provocative (and new at least to the referee).
  2. Evidence is presented of the correctness of this conjecture by analysis of models at weak coupling by formulating them on small spheres.
  3. The analogy to BEC suggest that methods used for the latter may be useful for understanding confinement in gauge theories.

We thank the referee for these positive and encouraging comments.

[Weaknesses]

  1. The models that can be treated this way require that the physics at weak coupling be continuously connected to strong coupling. While this may be true for certain supersymmetric theories it may not be true for other more physically relevant theories.

We certainly agree with the referee regarding this point. Nonetheless, we would like to note that our observations have made us cautiously optimistic. The close relationship between the Polyakov loop and ODLRO that we have discovered will allow one to explore the relationship between confinement and BEC at intermediate and strong coupling. Since in the latter case, the BEC at vanishing coupling and superfluidity at strong coupling can be connected without major subtleties, we deem it possible by our analogy that the same can be said for gauge systems.

  1. This analysis doesn't really lead to any new results that have not already been obtained using other methods. So the main point of the paper seems to be that it offers a new conceptual angle.

If the referee is referring to new computational results regarding the phase structure, we agree. We do however believe that we offered the physical mechanism to explain previously obtained computational results, and that a detailed understanding of this mechanism will be key to connect weakly-coupled and strongly-coupled regions.

[Report]

I think the paper offers some new and worthwhile observations and conclusions that are most likely relevant to the case of theories exhibiting gauge/gravity duality. I would have liked to see more discussion of eq. 1 - what is U that appears there? I understand that integrating over all gauge transformations produces a singlet but there seems to be more content than that in this eqn.

In the revised manuscript, we have provided some additional detail to Eq.(1) in order to explain the definitions of U and {\cal U} more clearly. In short, the operator {\cal U} acts on the states in the Hilbert space as the gauge transformation corresponding to the group element U.

Can one extend this analysis to finite (small) N by exploiting large N volume independence ? Eguchi-Kawai reduction should hold for many of the theories that are potentially being considered eg N=4 SYM, models with fermions in the adjoint representation etc. It would be nice for the authors to also include a brief discussion of these issues.

The following comment on page 22 in the discussion section is related to the referee's remark:

"Given that theories at small volume and large $N$ are often quantitatively close to those at large volume and moderate $N$ \cite{Eguchi:1982nm,GonzalezArroyo:1982hz}, it seems imaginable to also interpret confinement at finite $N$ as BEC."

In our revision, we have expanded on this comment and included several new references, so that the connection to the large-N volume independence becomes clearer.

Anonymous on 2020-07-23
(in reply to Nico Wintergerst on 2020-07-20)
Category:
remark

The authors have made a satisfactory response to my questions.

---

## Round 2 · Referee Report · Anonymous · 2020-6-7

Strengths

1. The paper is aimed at an interesting issue: color confinement remains an interesting problem, and a persuasive new connection between color confinement and Bose-Einstein condensation would be nice to understand.

Report

I was originally excited to read this paper because it claimed to make
an interesting new connection between Bose-Einstein condensation and
confinement-deconfinement transitions in gauge theory. That's a big
claim, but it is not supported by a reading of the manuscript. I
expected the manuscript to start by explaining why the claimed
relation is somewhat surprising, and then explain why it works anyway
- or least to have a discussion along these lines somewhere in the
paper. But it isn't there.

Bose-Einstein condensation takes place *only* in the infinite volume
limit, and requires a small but non-vanishing repulsive coupling. The
most common examples feature a U(1) global symmetry (and indeed, the
non-relativistic example of Bose-Einstein condensation in the
manuscript has a U(1) particle number symmetry), which breaks
spontaneously. Then the condensed phase is gapless, with a
Nambu-Goldstone boson.

4d SU(N) YM theory has a Z_N center symmetry which breaks
spontaneously in the deconfined phase when the spatial volume is
infinite. Both the confined and deconfined phases are always gapped.
(In the deconfined phase, the electric sector is gapped thanks to a
Debye mechanism, while the magnetic sector is gapped
non-perturbatively.) When N is sent to infinity, the Z_N symmetry
does not act like a U(1) symmetry, and there is no gapless
Nambu-Goldstone boson in the deconfined phase. When N is strictly
infinite, the confinement-deconfinement phase can survive in finite
spatial volume - this is the regime studied in the manuscript. But as
soon as N is finite, this phase transition ceases to exist at finite
volume, and becomes a smooth crossover.

These are all completely obvious differences between confinement
physics and Bose-Einstein condensation physics, and any proposal to
connect the two pictures should have addressed why they don't matter
right on page one, metaphorically speaking. Yet they are not
discussed at all.

Perhaps what the authors have in mind is drawing a connection between
Bose-Einstein condensation in scalar models in a 3d spacetime Z_N
symmetry, and the temperature-driven confinement-deconfinement
transition in 4d gauge theory, all in *infinite* volume. Then their
example in Section 2.3 is quite misleading, given that it has a U(1)
particle-number symmetry! But if this is the goal, then they would be
addressing a topic on which there is a huge amount of existing work
starting with the famous paper by Svetitsky and Yaffe,
http://old.inspirehep.net/search?p=recid:177233&of=hd . The
Svetitsky-Yaffe connection between symmetry breaking in a scalar field
theory and gauge theory is very direct, obviously works at finite N,
etc - so how is the proposal of the present manuscript an improvement
-- what does it add? But the Svetitsky-Yaffe paper isn't even cited,
so there are no comments on this!

Instead there are lots of arguments by analogy inspired by
manipulations of free theories and loosely worded claims. Just to
give two examples:

* Eq. 32 is claimed to be the "order parameter of the partial
deconfinement". But how precisely does its behavior distinguish
bona-fide distinct phases of matter? Do non-analyticities in the
quantity in Eq. 32 correspond to genuine phase transitions? The
paragraph below 32 gives a proposal for how it can be calculated
in general, and it appears to be a quite non-local quantity, so it
isn't obvious whether non-analyticities in it map to anything
sharp in terms of phase transitions. Is it at all meaningful at
finite N, given that Eq. 32 was obtained by studying a theory in
finite volume, where there are no phase transitions at all at
finite N? None of these questions are given a sharp answer, or
even acknowledged as serious concerns - at the end of the paper
there is some handwaving about how to extend the discussion to
finite N, but there is nothing sharp.

* Footnote 8 says "In the case of partial confinement, even if the
transition is not first order, the confined and deconfined phases
can coexist[14,11,15]. This happens for example if one introduce
fundamental matter[15]." It's hard to say what this is supposed to
mean. When fundamental matter is introduced with N_f/N \ll 1,
there is still a first-order phase transition as a function of
temperature. When N_f/N \sim O(1), this phase transition
disappears, unless it is forced to persist by the behavior of e.g.
chiral symmetry. So what is phase transition referred to in that
footnote?

To summarize, I do not think the manuscript makes a persuasive case
that its findings are sufficiently sharp and novel to be considered
for publication in SciPost.

  • validity: low
  • significance: low
  • originality: ok
  • clarity: low
  • formatting: good
  • grammar: reasonable

Author Nico Wintergerst on 2020-07-20
(in reply to Report 2 on 2020-06-07)

We thank the referee for their review of our work. However, with all due respect, while we appreciate the level of detail of the report, it appears in large parts to address what the referee thought we should have considered and written, but less so what we have actually tried to achieve with this paper.

We have clearly stated the scope of our work, providing detailed justifications for our definition of confinement and the reason why we work in the context of free theories. Moreover, we have openly pointed out the subtleties that emerge when extrapolating our findings to finite/strong coupling, providing some high level outlook on how our approach may leverage such an extrapolation but carefully avoiding strong statements in this direction. We are therefore rather surprised by the misunderstandings that lie at the root of this report.

In the following, we nevertheless attempt to address those points of the report that can be extrapolated to our work.

Bose-Einstein condensation takes place only in the infinite volume limit, and requires a small but non-vanishing repulsive coupling. .... ........... ........... These are all completely obvious differences between confinement physics and Bose-Einstein condensation physics, and any proposal to connect the two pictures should have addressed why they don't matter right on page one, metaphorically speaking. Yet they are not discussed at all.

As we have explained repeatedly throughout the manuscript, we are exploring the conceptual basis of our newly discovered relation between BEC and confinement, which is clearly stated as points 1.-4. in the introduction section of our manuscript. These basic points precede the details such as the gauge group, field content, representation, the order of transition, or the details of the energy spectrum (including existence/absence of an energy gap). A useful analogy, as explained in our paper, is the relation between Bose-Einstein condensation of non-interacting bosons and superfluid Helium. Although even the order of phase transition can be different, the essence of superfluidity is captured by the system of non-interacting bosons.

Regarding the comment, "Bose-Einstein condensation takes place only in the infinite volume limit": As a proper reading of the manuscript (in particular Sect. 2.3) would have easily exposed, we treat the system of N identical harmonically trapped bosons in the first quantized formulation, i. e. as a quantum field theory in 1+0 dimensions. In this sense, the system has zero-volume. Furthermore, what the referee refers to as the "infinite volume limit", also known as the thermodynamic limit, is realized as a large N limit, where N refers to the number of "fields" x_i, in this formulation. Thus, BEC and the deconfinement/confinement transition in large N YM theory on a finite volume are completely parallel, as we have carefully explained in section 2 of our manuscript. We wish to add that treating identical bosons and its condensation in the first quantized formulation is a standard way of dealing with this system.

Let us further highlight that we are characterizing the BEC using the permutation symmetry of the system, rather than the (spontaneously broken) U(1) global symmetry, as is also explained repeatedly in the manuscript. The understanding of the condensed phase from the permutation symmetry point of view is not only quite standard (as exemplified by the works in particular of Feynman and Penrose-Onsager, as explained in the text) but is also arguably of more fundamental nature than the U(1) symmetry. Namely, the permutation symmetry underlies the idea of off-diagonal longe range order (ODLRO), which, as is well-known, gives a microscopic understanding of the presence of the non-zero "macroscopic wave-function". Thus the permutation symmetry point of view (or the ODLRO) makes clear why and how the U(1) symmetry is broken in the condensed phase.

Perhaps what the authors have in mind is drawing a connection between Bose-Einstein condensation in scalar models in a 3d spacetime Z_N symmetry, ........ ........ so there are no comments on this!

In our manuscript we are not studying "scalar models in a 3d spacetime Z_N symmetry", and thus all the criticism raised by the referee in this paragraph is unwarranted. In fact, it is rather difficult for us to understand the possible origin of such a misunderstanding, given that we have clearly stated that we are dealing with the standard textbook example of the system of identical bosons already in the abstract. It should also be obvious from the table of contents which includes Sec. 2.3, titled "BEC of non-interacting particles" Sec. 4.2, titled "Polyakov loop for identical bosons".

  • Eq. 32 is claimed to be the "order parameter of the partial deconfinement". But how precisely does its behavior distinguish bona-fide distinct phases of matter?........ ........ ........ finite N, but there is nothing sharp.

Eq.(32) is a statement regarding the phases of the eigenvalues of the Polyakov loop. The Polyakov loop can be defined at each spatial point and is "local" in this sense. The use of the phases of the Polyakov loop in the manuscript is well-known to yield a sharp definition of distinct phases of large N gauge theory. As laid out in great detail in the manuscript, examples include the Gross-Witten-Wadia transition (Refs. 18, 19) and the Hagedorn transition for large N YM theories, as for example discussed in the important papers by Sundborg and Aharony et al. (Refs. 6, 7, 9).

  • Footnote 8 says "In the case of partial confinement, even if the transition is not first order, the confined and deconfined phases can coexist[14,11,15]. This happens for example if one introduce fundamental matter[15]."....... ................ ................ chiral symmetry. So what is phase transition referred to in that footnote?

We believe that from the context around Footnote 8, (Footnote 9 in the revised manuscript), and the footnote itself, it is evident that we are discussing the Gross-Witten-Wadia transition (or the Hagedorn transition) for YM theories coupled for example to fundamental matter fields. This would have been also clear by a casual glance at reference [15] cited here.

---

## Round 3 · Referee Report · Anonymous · 2020-7-23

Report

I don't have very much to add to my original report. I tried to make
a constructive suggestion about relating deconfinement and BEC of a
scalar field by bringing up the Svetitsky-Yaffe results, but it sounds
like the authors don't agree that there's any relation to what they're
saying. That's fine. But I still don't understand how to make sense
of their claims.

In their reply, the authors seem to suggest that all they want to do
is draw an analogy between BEC transitions in the thermodynamic limit,
and the deconfimenent transition of infinite-N gauge theory in finite
volume. (The manuscript speculates about finite-N extensions of the
results...)

I am certainly aware that the large N limit is a thermodynamic limit.
But in large N gauge theory the deconfinment phase transition is
always very strongly first order. But BEC transitions could be
either first or second order. The fact *even the order* of the
transition is doesn't agree between deconfinement and BEC is a loud
signal that any analogy between the systems is being pushed too far.

Moreover, I have to insist that the essense of superfluidity is the
existence of a gapless Nambu-Goldstone boson excitation in the
superfluid phase. It arises because of the spontaneous breaking of
U(1) particle number symmetry. The NGB is responsible for almost all
of the interesting phenomenology of superfluids. There's no such NGB
in gauge theory, either in the confined or deconfined phases.
Confinement famously comes with a mass gap (except when continuous
global symmetries unconnected to confinement are spontaneously
broken.) This is again a loud signal that the analogy the authors try
to draw is being pushed way too far.

  • validity: -
  • significance: -
  • originality: -
  • clarity: -
  • formatting: -
  • grammar: -

Author Nico Wintergerst on 2020-08-02
(in reply to Report 1 on 2020-07-23)

We thank the referee for the very prompt reply. Again, however, we notice that the report is rather tangential to the main message of our work. As we have explained multiple times, we discuss fundamental common aspects of the BEC and the deconfinement/confinement transition that precede model dependent details of the theory, including the order of the phase transition and the U(1) symmetry breaking. These main points of our work are drawn into focus repeatedly throughout our manuscript, yet, remarkably, they remain without direct commentary from the referee.

Let us nonetheless address below the objections raised in the referee's second reply.

(1)

I tried to make a constructive suggestion about relating deconfinement and BEC of a scalar field by bringing up the Svetitsky-Yaffe results, but it sounds like the authors don't agree that there's any relation to what they're saying. That's fine. But I still don't understand how to make sense of their claims.

The paragraph in the original report was:

Perhaps what the authors have in mind is drawing a connection between Bose-Einstein condensation in scalar models in a 3d spacetime Z_N symmetry, and the temperature-driven confinement-deconfinement transition in 4d gauge theory, all in infinite volume. Then their example in Section 2.3 is quite misleading, given that it has a U(1) particle-number symmetry! But if this is the goal, then they would be addressing a topic on which there is a huge amount of existing work starting with the famous paper by Svetitsky and Yaffe, http://old.inspirehep.net/search?p=recid:177233&of=hd . The Svetitsky-Yaffe connection between symmetry breaking in a scalar field theory and gauge theory is very direct, obviously works at finite N, etc - so how is the proposal of the present manuscript an improvement -- what does it add? But the Svetitsky-Yaffe paper isn't even cited, so there are no comments on this!

We are happy to learn this was a constructive suggestion rather than criticism. As is acknowledged by the referee now, we think that the main point of our work is unrelated to the results mentioned in this paragraph.

(2)

But in large N gauge theory the deconfinment phase transition is always very strongly first order. But BEC transitions could be either first or second order. The fact even the order of the transition is doesn't agree between deconfinement and BEC is a loud signal that any analogy between the systems is being pushed too far.

The criticism in this paragraph is unfounded in two ways.

First, it contains a simple factual error. The deconfinement phase transition is not always very strongly first order. For example, as the referee themself raised in the original report, with many fundamental fermions the transition is not of first order. For reference, see [15].

Second, as we have emphasized both in the manuscript itself and in our first reply, BEC and the deconfinement/confinement transition are characterized by fundamental properties which are independent of the order of the phase transitions. As we pointed out, the most fundamental example is the comparison between the lambda transition of superfluid He4, which is second order and the ideal Bose gas, which is third order. Would the referee challenge such an established idea as the analogy between the ideal BEC and the lambda transition of He4 based on the difference in the order of the transition?

(3)

Moreover, I have to insist that the essense of superfluidity is the existence of a gapless Nambu-Goldstone boson excitation in the superfluid phase. It arises because of the spontaneous breaking of U(1) particle number symmetry. The NGB is responsible for almost all of the interesting phenomenology of superfluids. There's no such NGB in gauge theory, either in the confined or deconfined phases. Confinement famously comes with a mass gap (except when continuous global symmetries unconnected to confinement are spontaneously broken.) This is again a loud signal that the analogy the authors try to draw is being pushed way too far.

Again, we are addressing fundamental aspects of BEC and the deconfinement/confinement transition that can be phrased independently of Nambu-Goldstone bosons. In fact, the language of spontaneous symmetry breaking is not particularly useful here. Let us remind the referee that on one side of the analogy, N is played by the number of particles, whereas on the other it corresponds to the rank of the gauge group. Consequently, we are forced to consider BEC in particle number eigenstates with fixed N and thus adopt the more useful framework of first quantization. As pointed out in the first reply, this framework allows one to characterize all of the condensation phenomena via the permutation symmetry, or ODLRO, and can arguably be considered more fundamental than the U(1) symmetry breaking. We may add here that, for example, a modern standard textbook on BEC, "Quantum Liquids: Bose condensation and Cooper pairing in condensed-matter systems" by A. Leggett employs this point of view throughout, and discusses all properties of BEC solely based on the ODLRO and the permutation symmetry.

We also wish to add that in gauge theories, whether a mass gap is formed is a model-dependent property (such as the matter content), as also acknowledged by the referee. Indeed, for a YM theory with gauge/gravity duality, gapless gravitons should exist. In the last section of our paper, we have in fact spelled out a possible analogy between the gapless mode dual to the graviton in YM/gravity theory and the NG boson (the phonon) in BEC.

---

## Round 3 · List of Changes

- Expanded discussion after Eq.(1)
- Expanded discussion of large-N volume independence on pg. 22
- Typos corrected in Appendix A

---

## Editorial Decision

editor-in-charge_assigned